

# Effects of motor imagery training on skeletal muscle contractile properties in sports science students

Björn Wieland[1], Michael Behringer[2] and Karen Zentgraf[1]

[1] Goethe University Frankfurt, Department of Sports Sciences, Movement and Exercise Science in Sports Unit, Germany
[2] Goethe University Frankfurt, Department of Sports Sciences, Sports Medicine and Exercise Physiology Unit, Germany

## ABSTRACT

**Background.** Studies on motor imagery (MI) practice based on different designs and training protocols have reported changes in maximal voluntary contraction (MVC) strength. However, to date, there is a lack of information on the effects of MI training on contractile properties of the trained muscle.

**Methods.** Forty-five physically active sport science students (21 female) were investigated who trained three times per week over a 4-week period in one of three groups: An MI group conducted MI practice of maximal isometric contraction of the biceps brachii; a physical exercise (PE) group physically practiced maximal isometric contractions of the biceps brachii in a biceps curling machine; and a visual imagery (VI) group performed VI training of a landscape. A MVC test of the arm flexors was performed in a biceps curling machine before and after 4 weeks of training. The muscular properties of the biceps brachii were also tested with tensiomyography measurements (TMG).

**Results.** Results showed an interaction effect between time and group for MVC ($p = 0.027$, $\eta^2 = 0.17$), with a higher MVC value in the PE group ($\Delta 5.9\%$) compared to the VI group ($\Delta -1.3\%$) ($p = 0.013$). MVC did not change significantly in the MI group ($\Delta 2.1\%$). Analysis of muscle contractility via TMG did not show any interaction effects neither for maximal radial displacement ($p = 0.394$, $\eta^2 = 0.05$), delay time ($p = 0.79$, $\eta^2 = 0.01$) nor contraction velocity ($p = 0.71$, $\eta^2 = 0.02$).

**Conclusion.** In spite of MVC-related changes in the PE group due to the interventions, TMG measurements were not sensitive enough to detect concomitant neuronal changes related to contractile properties.

## INTRODUCTION

Motor imagery (MI) is an internal simulation of movements without corresponding motor output (*Jeannerod, 1994*). Despite the absence of motor output and the absence of muscle activation measured by electromyography (EMG) (*Cowley, Clark & Ploutz-Snyder, 2008*; *Bunno & Suzuki, 2021*), improvements after repeated MI can be seen in various fields. For instance, the use of MI is discussed in human space flight to maintain muscle

Corresponding author
Björn Wieland, wieland@sport.uni-frankfurt.de

function (*Guillot & Debarnot, 2019*). More often MI is used in rehabilitation, *e.g.*, after stroke (*Monteiro et al., 2021*) or as support for motor learning (*Behrendt et al., 2021*). Furthermore, beside developing sport skills like swing and putting performance in golf due to MI (*Guillot et al., 2022*), another noteworthy effect is the increase in muscular strength (*e.g.*, *Yue & Cole, 1992*; *Bouguetoch, Martin & Grosprêtre, 2021*). These strength increases have been reported in studies using different designs and training protocols (*Paravlic et al., 2018*). The cited review suggests moderate beneficial effects of MI on maximal voluntary contraction (MVC) compared to no-exercise control groups and small beneficial effects compared to physical practice. Even if the strength increases are not as large as those following physical practice, they can still be considered as a substitute or additional training tool to maintain muscle function without subjecting athletes to maximum training intensities. While experts can be assumed to have a more vivid imagination (*Dhouibi et al., 2021*), there remains smaller room for adaptation in response to MI training, as trained individuals have a higher baseline level than untrained subjects (*Paravlic et al., 2018*).

However, although strength gains have been frequently observed after MI, it is still not clear how far these behavioral changes can be explained by adaptations on the cortical, spinal, or muscular level. Functional imagery studies have shown overlapping activation patterns between MI and actual movement (*Munzert, Lorey & Zentgraf, 2009*; *Wang et al., 2022*). Furthermore, on the corticospinal level, there is an increased excitability during MI compared to rest (*Mouthon et al., 2015*; *Yoxon & Welsh, 2020*). On the spinal level, results are more controversial looking at H-reflex (*Grosprêtre, Ruffino & Lebon, 2016*). However, it appears that subthreshold signals reach the spinal network and affect sensitive structures with a lower excitability threshold than alpha-motoneurons, *i.e.*, spinal interneurons (*Grosprêtre et al., 2019*). These subliminal signals also influence the muscular level, resulting in a shorter delay time of the muscle and hence an earlier onset of muscle contraction (*Wieland, Behringer & Zentgraf, 2022*). The above-mentioned effects of MI on the cortical, spinal, or muscular level are acute effects and have been investigated in imagery studies so far. However, long-term effects after regular MI sessions also need to be studied in order to better understand the mechanisms behind MI.

Although there have been a large number of behavioral studies after some level of MI practice (*Yue & Cole, 1992*; *Bouguetoch, Martin & Grosprêtre, 2021*), there are fewer studies addressing cortical, neuronal, or muscular adaptations. In a review paper, *Ruffino, Papaxanthis & Lebon (2017)* showed that improvements in motor behavior (including strength performance) following MI practice are accompanied by a reorganization of cortical structures. In terms of cortical mapping, they found an increased representation of the trained muscles and increased corticospinal excitability in the first weeks of practice. *Hale, Raglin & Koceja (2003)* hypothesized that effects of MI on spinal excitability would be more pronounced after repetitive MI practice. This premise was supported by *Grosprêtre et al. (2019)*, who showed short-term plasticity of the spinal network induced by repetitive MI practice. Nonetheless, both studies looked only at short-term adaptations. Results of studies in which MI training was performed for 1 and 2 weeks, respectively, showed an increase in MVC and rate of torque development (RTD) in addition to an increase in V-wave during MVC without associated changes in the superimposed H-reflex. This

suggests neural adaptations on the supraspinal level (*Grosprêtre et al., 2018*; *Bouguetoch, Martin & Grosprêtre, 2021*). Another indication of supraspinal adaptions through MI training is a significant increase in strength of the contralateral side (*Yue & Cole, 1992*; *Bouguetoch, Martin & Grosprêtre, 2021*). The underlying mechanisms may be cortical reorganization leading to a higher central command or better coordination of muscle activation (*Ranganathan et al., 2004*).

Recent literature also provides evidence of spinal adaptation. For example, an increased H-reflex at rest was observed after regular MI practice over multiple sessions (*Grosprêtre et al., 2018*; *Bouguetoch, Martin & Grosprêtre, 2021*). These authors suggest that MI results in subthreshold cortical motor output that modulates sensitive spinal structures such as interneurons, and that this leads to a reduction of spinal presynaptic inhibition. This reduction of presynaptic inhibition, while afferent-motoneuronal synapses remained unchanged, could be shown after an acute bout of MI (*Grosprêtre et al., 2019*).

There is widespread agreement that a first-person perspective during MI is more likely to result in peripheral physiological effects than a third-person perspective (*Yue & Cole, 1992*; *Reiser, Büsch & Munzert, 2011*). Furthermore, instructions should be task-specific and close to the target movement. There is evidence that proprioceptive information about the actual body posture leads to stronger imagination. For example, there is an increased vividness (*Guilbert et al., 2021*) and greater motor-evoked potentials (MEP) when real and imagined hand movement are compatible (*Vargas et al., 2004*; *Lorey et al., 2009*). In a 6-week intervention study, *Jiang et al. (2017)* showed benefits of low contractions during MI practice compared to low contractions or imagery alone. For this reason, the present study uses an "Effort" condition during pre and post measurements that has already been used in previous research (*Wieland, Behringer & Zentgraf, 2022*).

The effects of regular MI training have rarely been studied on a muscular level. So far, anatomical characteristics have been described showing no changes in pennation angle or fascicle length after regular MI training (*Bouguetoch, Martin & Grosprêtre, 2021*). In the before-mentioned study, the authors also showed unchanged twitch responses after MI training and concluded that strength gains after MI training were solely due to neural plasticity.

Muscle contractility is not only characterized by cellular or structural aspects, but also by neuroplasticity and its influence on the muscle. Consequently, even in the absence of structural and cellular changes in the muscle, the kinematics of muscle contraction may be altered by neuronal effects. Potential changes in muscular contractility could be measured using tensiomyography (TMG). TMG is a non-invasive method examining the mechanical response of muscle contractions in response to electrical stimulation and is often used as a method for detecting muscular, mechanical and neuronal properties and changes, also fatigue-related (*García-Manso et al., 2012*; *Hunter et al., 2012*; *Wilson et al., 2019*; *Zubac & Šimunič, 2017*). TMG can be used to obtain maximal radial displacement ($D_m$), contraction velocity ($V_c$), and delay time ($T_d$) of the electrically stimulated contraction, and these parameters can be used to draw different conclusions. *Macgregor et al. (2018)* summarize the results of TMG studies in their review article and interpret these parameters as follows: $D_m$ reflects muscle belly stiffness, which decreases after regular strength training, and is

accompanied by an increase of maximal strength. $V_c$ can be used to determine information about contraction velocity without an influence from changes in $D_m$. $T_d$ provides a measure of muscle responsiveness and shows the onset of muscle contraction. The mentioned TMG parameters are influenced by both structural and neuronal parameters. Thus, $D_m$ decreases due to greater muscle thickness (*Wilson et al., 2019*) or due to higher muscle tone (*Pišot et al., 2008*). $V_c$ is influenced by composition in fiber distribution (*Zubac & Šimunič, 2017*) and in activation level of the muscle (*García-Manso et al., 2012*). Hence, TMG is considered to detect early hallmarks of muscle contractility before overt architectural changes occur (*Šimunič et al., 2019*). Therefore, TMG seems to be a useful method to detect possible changes in the muscle contractility due to MI training even in the absence of structural or cellular adaptations.

The practice of the MI group included maximal isometric contractions of the biceps brachii over 4 weeks. Load parameters were based on the results of a meta-analysis (*Paravlic et al., 2018*). As a comparison, a physical exercise (PE) group performed the imagined movement physically in order to execute maximal isometric contractions of the biceps brachii in a biceps curling machine. A visual imagery (VI) group was used to simulate the cognitive load of the MI group, but on a different task. This VI group was instructed to generate imagery over the same period of time. However, to avoid imagery effects in this group, they had to imagine a landscape scenery instead of movement. With these three intervention groups and the used measurement method, it is possible to show changes in muscle contractility alongside associated strength gains, comparing three different interventions. To date, only structural changes or a change in twitch response at the muscular level have been studied for MI interventions; information on the kinematics of contraction, which is influenced by neuronal changes in addition to structural changes, is not yet available for MI interventions.

### Objectives

The aim of this study was to investigate changes of muscular strength & muscle-contractile properties after MI & PE practice to examine whether and to what extent cortical and spinal changes due MI or PE training are measurable at the muscular level. In addition to the change in strength, the kinematics of the contraction, *i.e.*, its onset, velocity, and maximal displacement, provide new information about possible adaptations. The hypothesis for the intervention is that both practice groups (MI & PE) improve in strength and show altered muscle contractility, with more pronounced changes in both parameters in the PE group. Lower $D_m$ and higher $V_c$ values due to increased cortical output (*Ranganathan et al., 2004*) and a lower $T_d$ due to reduced presynaptic inhibition (*Grosprêtre et al., 2019*) after regular MI practice are expected. Accordingly, the research question is whether repeated MI (& PE) training alters muscle contractility.

## MATERIALS & METHODS

### Procedure

The study was made public and promoted in the lectures of the department of sport science at the Goethe University. Participation was voluntary and each participant provided written

consent for participation in the study that was approved by the local ethics committee (Department of Psychology and Sports Science, Ethics committee, Ethical Application Ref: 2019-63). All measurements were carried out by the same investigator in the same location with constant environment at the Institute of Sports Sciences, Goethe University Frankfurt.

## Subjects

Of the 45 sport science students recruited (21 female, 24 male, $M_{age}$ 24.9 ± 3.7 years), 42 completed the intervention. One dropped out due to illness and two were excluded due to technical problems with the final measurement. Participants were matched on the basis of gender and MVC value and divided into three groups (MI: six female, seven male, $M_{age}$: 25.6 ± 4.7 years; PE: six female, eight male, $M_{age}$: 23.6 ± 2.0 years; VI: seven female, eight male, 26.2 ± 3.6 years). All participants had experience in strength training due to their academic background in sport science and their recreational sports activities. Experience with MI practice was sporadic; all participants completed a familiarization MI session. Further, the subjects were asked to immediately inform the researcher about any change in their lifestyles and daily activities. Exclusion criteria were injuries of the musculoskeletal system of the upper body during the last six months before the study. Participants with cardiopulmonary or neurological and orthopedic disorders were also excluded.

## Study design

A baseline measurement (pre) was performed on all participants by adjusting and documenting the sitting position on the biceps curl machine (Schnell, Germany) before measuring maximal voluntary contraction (MVC) strength. Furthermore, imagery and effort conditions (0 N, relaxed position, hands fixed on the biceps curling machine and 50 N, low contraction against the biceps curling machine to produce 50 N) were practiced in order to familiarize participants with TMG. From two to a maximum of seven days later, the muscular properties of the arm flexors were measured during MI of maximal voluntary contraction of the biceps brachii as a second part of the premeasurement. MI instructions were based on the PETTLEP model (Physical, Environment, Task, Timing, Learning, Emotion, Perspective) from *Holmes & Collins (2001)*. For TMG measurements, participants were seated and their wrist was fixed as determined at the first appointment. Measurements started with the 0 N condition in order to avoid fatigue. Before the 50 N condition, participants took a 3-min break. The instructions for the imagery task were repeated at the beginning of each condition. The same keywords ("get ready," "go," and "stop") were used during each measurement. As a check, participants were asked to rate the vividness of each trial using a five-point Likert scale based on the VMIQ-2 (*Zabicki et al., 2017*). Trials with a drop of two points or more on the vividness scale were repeated.

Intervention groups were matched based on gender and relative MVC at baseline measurement. The MI group conducted MI practice of maximal isometric contraction of the biceps brachii during first-person observation; the PE group physically practiced maximal isometric contractions in a biceps curling machine; and the VI group performed visual-imagery training of a landscape. Based on the results of a meta-analysis, a period of 4 weeks, a training frequency of three times per week and a session duration of about 15

min per session was associated with enhanced strength gains after MI training (*Paravlic et al., 2018*). For adequate comparability, all three groups followed identical time intervals according to the results of this meta-analysis. The participants in the three training groups practiced three times per week over a 4-week period. All three groups were active for the same amount of time (12 training sessions lasting 14 min per session). The three groups differed in the specific execution within a training session in order to design an effective training for each condition. Thus, the MI training was based on the results of the before mentioned meta-analysis and the load parameters of the PE group were based on conventional maximum strength training.

## Measurement

TMG measurements were performed as previously described in *Wieland, Behringer & Zentgraf (2022)*, which means that the displacement-measuring sensor was placed on the point of maximal muscle belly circumference detected by manual palpation during contraction of the biceps brachii muscle of the right arm. Self-adhesive electrodes (5 × 5 cm, axion GmbH, Leonberg, Germany) were placed symmetrically at a distance of 5 cm lateral and medial to the displacement-measuring sensor on the biceps brachii (*Wieland, Behringer & Zentgraf, 2022*). The stimulation, a single monophasic square wave with 1 ms pulse width, started with 20 mA and was increased by 10 mA every 30 s to minimize the effects of fatigue and potentiation (*Wilson, Johnson & Francis, 2018*). The intensity of stimulation was increased until the mechanical response of the muscle was maximal or 100 mA was reached. This protocol was first performed for the 0 N condition and after a 3 min break also for the 50 N condition. The TMG has been shown to be reliable for the biceps brachii muscle (*Krizaj, Simunic & Zagar, 2008*). Both the measuring point and the positions of the electrodes were marked with permanent ink to ensure identical measuring locations in consecutive measurements. During the 0 N and 50 N conditions, participants received live feedback via a force–time curve on a screen using Diagnos 2000 (Trainsoft GmbH, Moorenweis, Germany) connected to the biceps curl machine. The exclusion criterion for the TMG measurements during the 50 N trials was failure to maintain the force in the range of 40 to 60 N. For the 0 N condition, muscle activity was controlled using EMG. The EMG electrodes (Covidien, Kendall electrodes H93SG, two cm interelectrode distance) were placed according to the recommendations of the SENIAM initiative (Surface Electromyography for the Non-Invasive Assessment of Muscles), *i.e.*, the skin was shaved and dry-cleaned with alcohol to maintain low impedance. Electrodes were placed parallel to the muscle fibers proximal and distal to the TMG sensor on the biceps brachii (Fig. 1). The TMG and EMG setup was also used in *Wieland, Behringer & Zentgraf (2022)* where it is described in more detail.

   To evaluate elbow flexor strength, a static MVC test was performed in a biceps curl machine. The peak value of the force-time curve was calculated with the software Diagnos 2000 (Trainsoft GmbH, Moorenweis, Germany). Each participant had three attempts with a rest of 3 min in between, the attempt with maximum performance was selected for further analysis.

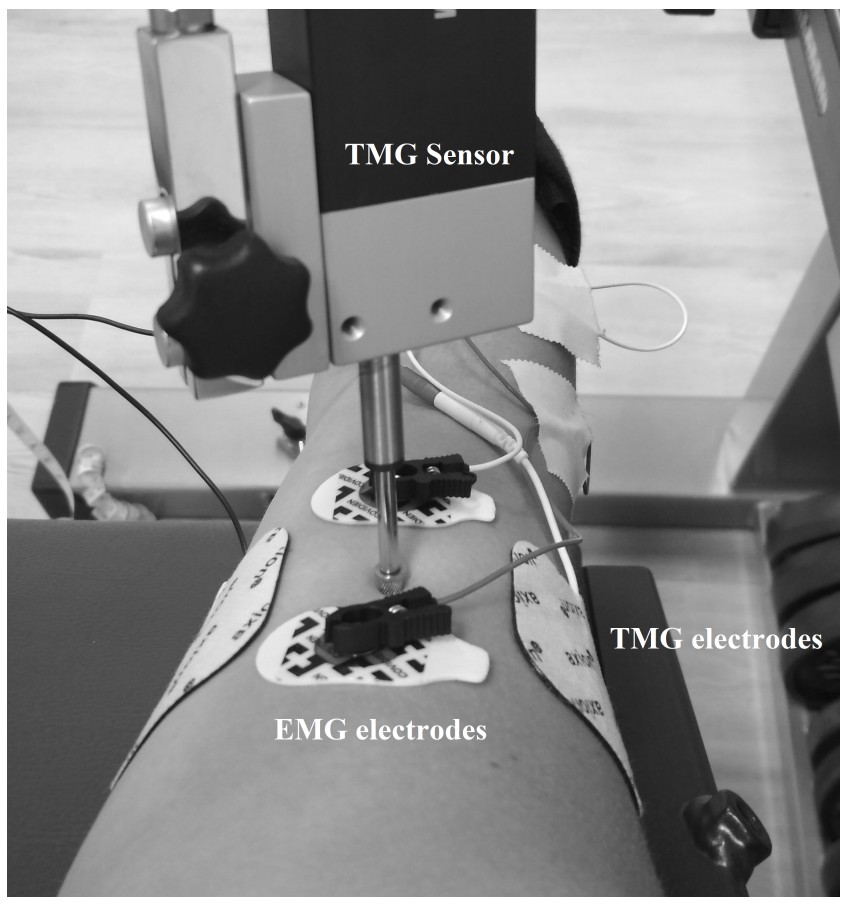

**Figure 1** Location of the TMG and EMG electrodes as well as the TMG sensor.

Imagery ability was determined using either the Vividness of Movement Imagery Questionnaire (VMIQ-2; *Roberts et al., 2008*) or its German translation (*Dahm et al., 2019*) depending on the native language.

## MI Training

Participants in the MI group were asked to practice MI of maximal isometric contraction of the biceps brachii during first-person observation with kinesthetic modality using PETTLEP-based instructions (*Holmes & Collins, 2001*). Parameters such as duration and number of repetitions were based on the results of *Paravlic et al.*'s meta-analysis (*2018*). Accordingly, participants in the MI group completed a 4-week training period with a frequency of three times per week. The training usually took place on Monday, Wednesday, and Friday between 8 am and 5 pm. Each session consisted of two sets with 25 repetitions per set. Each imagination was performed for 5 s with 10 s rest between each repetition and 90 s set rest. To avoid muscle contractions of the biceps brachii (*i.e.*, the participant's right arm) during MI, participants received feedback with the help of EMG.

## Physical exercise

The load parameter of the PE group was matched to the MI group in terms of number of sessions and duration of training. Therefore, the group also trained for 4 weeks with a frequency of three times per week, usually on Monday, Wednesday, and Friday between 8 am and 5 pm. Participants performed two 5 s maximal effort biceps curls in a biceps curling machine separated by a 15 s resting interval. They completed five sets per session with a 3 min set break.

## VI training

The VI group imagined a landscape while the instructor gave details about its environment. Instructions were chosen to prompt a third-person imagination without movement. Each of the four imagination sessions lasted 2.5 min with an 80 s break between sessions. During VI training sessions, the biceps brachii was controlled for involuntary muscle contractions using electromyography. Like the other two groups, the training took place over 4 weeks with three sessions (usually on Monday, Wednesday, and Friday between 8 am and 5 pm) per week.

## Data analysis

MVC was calculated using the software Diagnos 2000 and defined as the peak of the force–time curve during the maximal static contraction of the biceps brachii. Each individual's highest value was used for further analysis and divided by body mass (kg) to obtain a relative force value.

For the VMIQ-2 score, participants were asked to rate their individual ability to imagine themselves performing 12 simple motor tasks on a scale ranging from 1 (*perfectly clear and vivid*) to 5 (*no image at all, you only know that you are "thinking" of the skill*). Each item was summed to gain a vividness score for each component (internal visual imagery, external visual imagery, and kinesthetic imagery) of between 12 and 60, with lower scores indicating more vivid images (*Dahm et al., 2019*; *Wieland, Behringer & Zentgraf, 2022*).

All TMG variables were calculated using the maximal radial displacement curve over time. The highest point of the curve, the maximal radial displacement ($D_m$), is expressed in mm. The delay time ($T_d$) represents the time in ms between the electrical impulse and 10% of the maximal displacement. Both parameters were calculated by the TMG software. Contraction velocity ($V_c$) was calculated via MATLAB R2019b (MathWorks, USA). For this, the slope of the displacement curve over time was calculated using the following formula:

$$V_c = \frac{(90\% \ D_m - 10\% \ D_m)}{(\text{contraction time from 10\% to 90\% of } D_m)}.$$

For statistical analysis, only the trial with the highest value of $D_m$ for each condition and both measuring times were included.

EMG signals were acquired with a frequency of 5000 Hz by the Biopac System (MP160 BIOPAC EMG-2R wireless sensor) and filtered with a band-pass filter (bandwidth 12 to 500 Hz). The last 500 ms before external stimulation of the TMG stimulator was used to calculate the root-mean square EMG (RMS). Only those EMG data of the included TMG trials were analyzed.

## Statistics

Statistical analyses were performed using IBM SPSS (Version 26, IBM Corporation, Armonk, New York). Descriptive values are expressed as means ($M$) and standard deviations ($SD$).

A mixed-model analysis of variance (ANOVA) was used to determine effects of groups (MI, PE, VI) over time (Pre, Post) for MVC. Additionally, as in other MI studies (*Reiser, Büsch & Munzert, 2011*), average percentage gain in strength within a participant was used for statistical post hoc analysis.

To determine effects of muscle-effort conditions (0 N, 50 N), time (Pre, Post) and group (MI, PE, VI), a 2 × 2 × 3 ANOVA was used for TMG parameters. Effect size is reported as partial eta squared.

To control for muscle activation during measurements, the prestimulus background EMG activity was compared to the baseline measurement of the respective person. The RMS of EMG signals 500 ms before the TMG stimulus was checked for differences in mean values using a paired $t$ test.

The level of significance was set at $\alpha < 0.05$.

## RESULTS

Thirteen participants in the MI group (six female, seven male, $M_{age}$: 25.6 ± 4.7 years), 14 in the PE group (six female, eight male, $M_{age}$: 23.6 ± 2.0 years), and 15 in the VI group (seven female, eight male, $M_{age}$: 26.2 ± 3.6 years) completed the intervention. Overall VMIQ-2 score was 19.1 (±5.9) for MI, 24.1 (±5.1) for PE, and 20.8 (±5.6) for VI. There were no significant differences between pre and post EMG values in the 0 N condition $t(39) = -0.64$, $p = 0.526$.

### Strength

The results of the 2 × 3 (Time × Group) ANOVA showed an interaction effect for MVC, $F(2, 39) = 3.9$, $p = 0.027$, $\eta^2 = 0.17$, but no main effect Time, $F(1, 39) = 3.4$, $p = 0.073$, $\eta^2 = 0.08$ and no main effect Group, $F(2, 39) = 0.09$, $p = 0.913$, $\eta^2 = 0.005$ (Fig. 2). Post-hoc comparisons of the %-change of MVC (PE: 5.9%, MI: 2.1%, VI: −1.3%) with independent two sample $t$-tests revealed significantly higher improvements in the PE compared to the VI group, $t(27) = -2.65$, $p = 0.013$, but no significant differences between MI and PE, $t(25) = -1.45$, $p = 0.159$, as well as no significant differences between MI and VI, $t(26) = 1.83$, $p = 0.079$. Consequently, the interaction of the MVC values is due to the fact that the MVC of the PE group increases and that of the VI decreases.

### Skeletal muscle contractile properties

The 2 × 2 × 3 (Time × Effort × Group) ANOVA for maximal radial displacement of the biceps brachii revealed no significant Time × Effort X Group interaction, $F(2, 39) = 0.96$, $p = 0.394$, $\eta^2 = 0.05$, no main effect Time, $F(1, 39) = 0.45$, $p = 0.505$, $\eta^2 = 0.01$ or Group effect, $F(2, 39) = 1.05$, $p = 0.359$, $\eta^2 = 0.05$ but a main effect of Effort, $F(1, 39) = 1120$, $p < 0.001$, $\eta^2 = 0.97$ (Fig. 3). *Post hoc* analysis showed higher values for no muscle effort ($p < 0.001$). There are no group, time or interaction effects for maximal

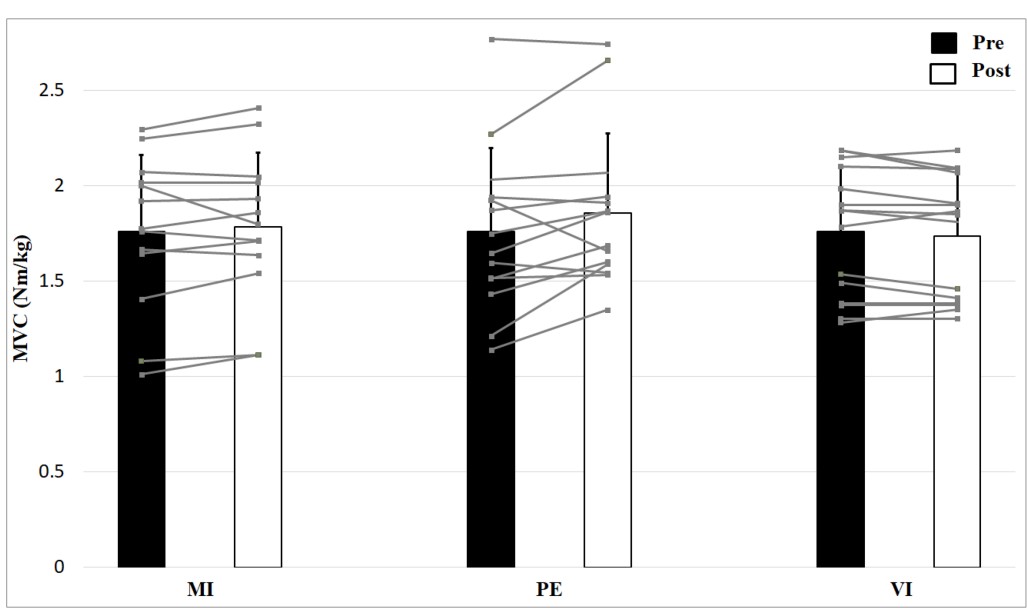

**Figure 2** Relative strength values (means + 1 standard error) for the three groups before (black bars) and after (white bars) intervention. Gray lines show individual values of the participants.

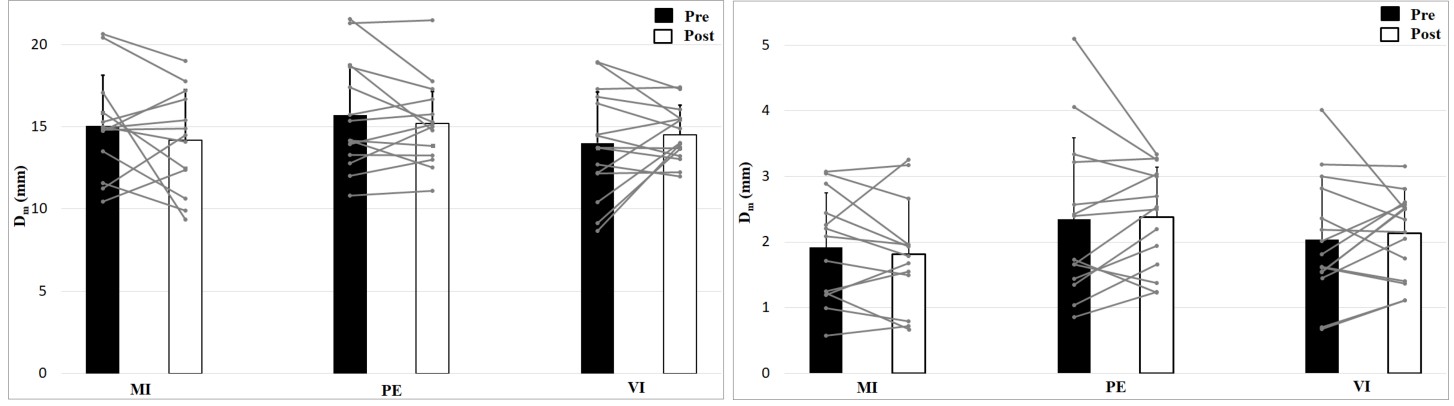

**Figure 3** Maximal radial displacement ($D_m$; means + 1 standard error) of the three groups before (black bars) and after (white bars) intervention. Gray lines show individual values of the participants. The left figure shows the 0 N, the right figure the 50 N effort condition.

radial displacement. For the 50 N condition there is a smaller displacement than for 0 N condition.

Analysis of contraction velocity revealed no significant interaction effect (Time × Effort × Group), $F_{(2, 39)} = 0.35$, $p = 0.71$, $\eta^2 = 0.02$, and no Time effect, $F_{(1, 39)} = 1.53$, $p = 0.22$, $\eta^2 = 0.04$, or Group effect, $F_{(2, 39)} = 0.50$, $p = 0.61$, $\eta^2 = 0.03$, but a main effect of Effort, $F_{(1, 39)} = 952$, $p < 0.001$, $\eta^2 = 0.96$, with higher values for 0 N (Fig. 4). Contraction velocity showed no differences between group or time as well as no interaction

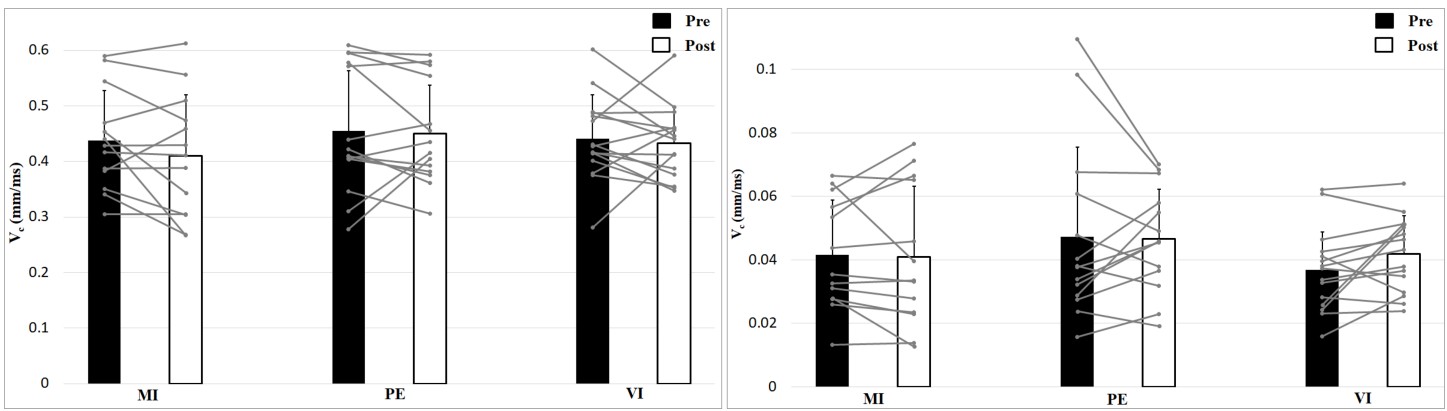

**Figure 4 Contraction velocity ($V_c$; means + 1 standard error) of the three groups before (black bars) and after (white bars) intervention. Gray lines show individual values of the participants.** The left figure shows the 0 N, the right figure the 50 N effort condition.

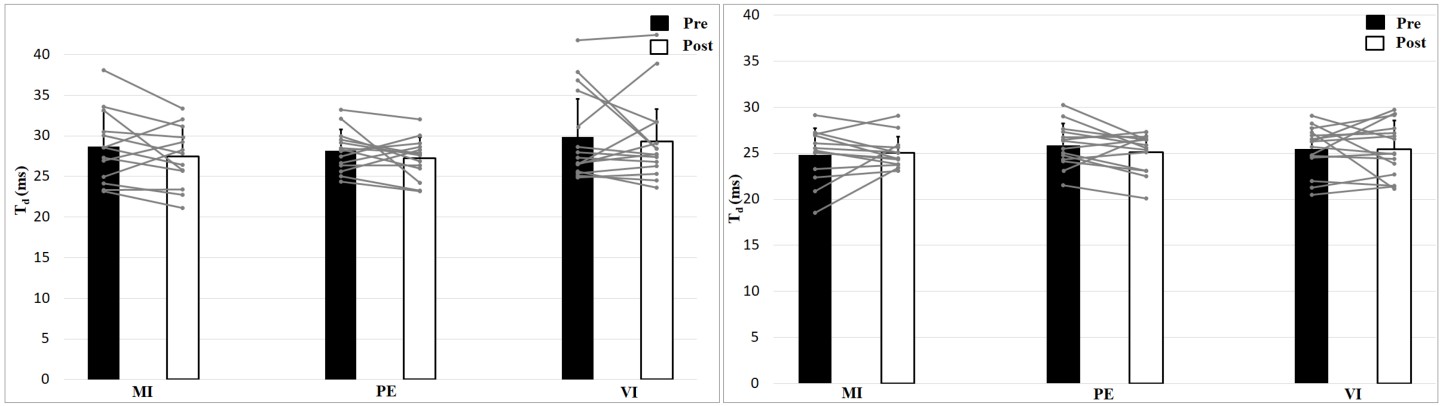

**Figure 5 Delay time ($T_d$; means + 1 standard error) of the three groups before (black bars) and after (white bars) intervention. Gray lines show individual values of the participants.** The left figure shows the 0 N, the right figure the 50 N effort condition.

effect. Contraction velocity is higher in the 0 N condition compared to the 50 N condition.

Analyzing delay time with a 2 × 2 × 3 (Time × Effort × Group) ANOVA showed no interaction effect, $F(2, 39) = 0.24$, $p = 0.79$, $\eta^2 = 0.01$, and no Time effect, $F(1, 39) = 1.16$, $p = 0.29$, $\eta^2 = 0.03$, or Group effect, $F(2, 39) = 1.11$, $p = 0.34$, $\eta^2 = 0.05$ (Fig. 5). Muscle Effort differed significantly with lower values for 50 N, $F(1, 39) = 30.7$, $p < 0.001$, $\eta^2 = 0.44$. Delay time is shorter in the 50 N compared to the 0 N condition. There are no differences between group, time and no interaction effect.

## DISCUSSION

The objective of this study was to investigate MVC changes in an isometric biceps curl task and whether contractile properties of the biceps brachii muscle are altered after regular MI or PE training. This is the first study looking at muscle contractile properties using

TMG measurements after a 4-week MI intervention. A positive effect in MVC strength was shown for the PE group (5.9%). No significant changes were found for MI or VI. Since VMIQ-2 scores indicate that participants' vividness of imagery was clear and reasonably vivid, a poor imagination does not seem to be the reason for the lack of strength gains due to MI. However, comparing the percentage change of MVC (done so in *Yue & Cole, 1992*; *Reiser, Büsch & Munzert, 2011*; *Grosprêtre et al., 2018*) reveals that the MI group increased by about one third of the size of the PE group (PE: 5.9%; MI: 2.1%). The VI group showed a small decline of MVC (−1.3%). Compared to similar studies, the MI (as well as the PE) effect observed in the present study is rather small. Other intervention studies found strength gains due to MI of about 5–30% (for an overview, *Paravlic et al., 2018*). One possible reason for the rather small MVC effects in this current study, which is underlined by the drop of strength in the control group (VI), could be that the intervention period took place while there were restrictions on social life such as recreational sports due to the COVID-19 pandemic. Participants seemed to be less physically active during the intervention period (*Zaccagni, Toselli & Barbieri, 2021*), and, therefore, to some extent could have experienced the effects of detraining. Moreover, all participants were sport science students at the Goethe University Frankfurt and in addition to their recreational sports, they had to participate in two to four sport classes (90 min each) during each academic period (except for remote learning during the COVID-19 pandemic). With this in mind, they can be considered to be trained participants and their capacity for adaptations due to MI training might be lower than in untrained populations (*Paravlic et al., 2018*). A further explanation of the comparably small strength gains is that training of the upper body showed smaller effects than training of the lower body (*Paravlic et al., 2018*). These arguments are also valid for the PE group, indicating that the training frequency and duration seemed to be too low for well-trained persons to obtain major effects.

Given the smaller increases in MVC, a smaller effect on contractility might also be expected. As the argument applies to MVC gains, stating that trained persons have less potential to adapt in strength, it also applies to TMG parameters. For instance, results indicate that muscles with lower habitual load show higher adjustments in $D_m$ and $T_c$ (*Zubac & Šimunič, 2017*). This can be confirmed by the analyses of the TMG parameters, which reveal no interaction effect and no main effect for time or group. Only a main effect of *effort* for all TMG parameters was found and can be explained by the voluntary contraction of the muscle and the corresponding change of muscle properties. Voluntary contraction occurs during the 50 N condition, in which a low force must be kept during the TMG measurement. The additional proprioceptive information should enhance the MI effects. The missing interaction effect between the effort condition and time and group indicates that the MI group had no additional benefit from the proprioceptive information compared to the PE and VI group. As there are also no group, time or group x time interaction effects, this shows that neither the PE nor the MI group had any significant change in contractility over time. Looking at the percentage changes, small tendencies in the TMG variables can be seen, $D_m$ (PE: −2.5%, MI: −5.7%, VI: 3.8%), $T_d$ (PE: −3.2%, MI: −1.8%, VI: −0.9%), $V_c$ (PE: −1.4%, MI: −6.3%, VI: −0.8%), which, however, do not show any significant differences. Accordingly, this study showed that maximal strength training

as well as MI does not lead to any changes in muscle contraction assessed by TMG. To the best of our knowledge, no study with long-term MI intervention used TMG measurements. The unchanged TMG parameters may indicate that there are no corticospinal adaptations caused by MI training that affect muscle contractility. However, there was also no change in the PE group. For this reason, the TMG parameters and the respective mechanisms are discussed in more detail in the following sections.

The expectations for the parameter $D_m$ was a reduction in the PE and MI groups due to a higher muscle stiffness associated with improved muscle function (*Zubac & Šimunič, 2017*; *Wilson et al., 2019*). Higher muscle stiffness was supposed to result from an increased cortical neural drive affecting the spinal motoneuronal pool (*Grosprêtre et al., 2018*). However, no changes in $D_m$ could be detected in either group. In studies using physical practice as intervention, a reduction of $D_m$ was shown after 8 weeks of plyometric training (*Zubac & Šimunič, 2017*), 6 weeks resistance training (*Wilson et al., 2019*) or 7 weeks resistance training (*Kojić et al., 2021*; *Kojić et al., 2022*), together with an increase in jumping performance or strength. As far as the mentioned studies also surveyed muscle thickness, an increase was also shown for this parameter alongside the decrease of $D_m$ (*Kojić et al., 2021*; *Kojić et al., 2022*; *Wilson et al., 2019*). The results of *Pišot et al. (2008)* confirm these findings: They showed an increase in $D_m$ along with a decrease in muscle thickness during a 35-day bed rest study. Additionally, the studies of *Pišot et al. (2008)* and *Kojić et al. (2021)* correlated change of muscle thickness and change of $D_m$, showing a high relationship of these two variables ($r = -0.70$, $p < 0.01$ and $r = -0.76$, $p < 0.01$). This suggests that participants with a greater increase in muscle belly thickness experienced greater decreases in $D_m$ (*Kojić et al., 2021*), or a greater decrease in muscle belly thickness was associated with a greater increase in $D_m$ (*Pišot et al., 2008*). *Kojić et al. (2022)* also observed this relationship suggesting that a reduced $D_m$ value reflects muscle hypertrophy ($r = -0.72$, $p < 0.01$ and $r = -0.76$, $p < 0.01$), but moreover revealed the absence of correlations between strength gains and the corresponding TMG changes ($r = -0.13$, $p = 0.72$ and $r = -0.35$, $p = 0.32$). This might indicate that changes in $D_m$ following resistance training are more influenced by gains in muscle thickness rather than by stiffness, which could explain the lack of effects of $D_m$ in this study. The training design of the PE group was different from the above-mentioned interventions: maximal contractions (rather than submaximal) were performed with a low number of repetitions. According to *Zatsiorsky & Kraemer (2006)*, one would classify this intervention as *maximal effort method*, in which neuronal rather than hypertrophy effects are expected. Also, MI interventions are not expected to result in muscle thickness growth (*Bouguetoch, Martin & Grosprêtre, 2021*) which likewise could explain the absence of effects for $D_m$ in this group. Furthermore, 4 weeks of resistance training are rather short for hypertrophy effects and mainly neuronal adaptations can be expected (*Sale, 2003*).

Contraction velocity between power and endurance athletes differ, with lower contraction times for power athletes (*Loturco et al., 2015*; *Šimunič et al., 2018*), consistent with the correlation between proportions of myosin heavy chain I and contraction time (*Šimunič et al., 2011*). Through temporary training intervention, *Zubac & Šimunič (2017)* showed a decrease in contraction time due to plyometric training, but this could not be

shown for resistance training (*Kojić et al., 2021*; *Wilson et al., 2019*). The absence of an effect on $T_c$ after resistance training could be due the dependence on $D_m$. Therefore, in this study, $V_c$ was chosen as measure of contraction velocity as it is independent from $D_m$. In addition to a change in fiber distribution (*Zubac & Šimunič, 2017*), a modification of the contraction velocity can also be achieved by a changed activation level of the muscle (*García-Manso et al., 2012*). As MI is not thought to cause any fiber change, a modified activation level through an increased cortical drive after regular MI training (*Ranganathan et al., 2004*) could cause a change in $V_c$. However, increased contraction velocity through PE or MI was not shown in this study, which may be due to either the type (resistance and MI instead of explosive strength training) and/or (short) duration of the intervention.

*Toskić et al. (2022)* showed a higher muscle responsiveness, meaning lower values for $T_d$, in physically active compared to physically inactive people. However, the parameter delay time has rarely been investigated in intervention studies. The hypothesized reduced presynaptic inhibition after regular MI practice (*Grosprêtre et al., 2019*) or resistance training (*Aagaard et al., 2002*) would lead to a decrease of $T_d$, which has already been shown for acute MI effects (*Wieland, Behringer & Zentgraf, 2022*). Concerning the previously discussed TMG parameters, no group or time effect could be shown for delay time.

## CONCLUSIONS

The purpose of this study was to determine whether cerebral or spinal adaptations due to MI or PE training can also be observed on the muscular level and whether they result in strength gains and altered muscle contractility. Strength gains were only observed for the PE group. For MI, *Bouguetoch, Martin & Grosprêtre (2021)* already showed no changes in pennation angle, fascicle length and an unchanged twitch response. In the current study, no changes in muscle contractility measured by TMG were observed for both groups, MI and PE. As no effects on the muscle could be shown for the PE group either, further studies measuring the effects of neuronal changes on the muscle would be important, especially for MI.

## LIMITATIONS AND FUTURE PERSPECTIVES

In conclusion, no significant changes for muscular properties after each intervention and between all three groups were observed. As discussed, one reason for the lack of effects could be that the training was neither designed for thickness growth (see $D_m$) nor for explosive strength (see $V_c$) and maybe the duration of the intervention was too short for these kinds of adaptations. Another reason for the small effects could also be the external circumstances (COVID-19, trained participants, upper body musculature), as shown by the small effects in MVC. Although the observed MVC effects are smaller than in other studies, the pattern of strength gains of PE and MI are comparable to the results of other interventions (*Paravlic et al., 2018*). Therefore, future studies should include a longer duration of the intervention and, at least for the PE group, repetitions and intensity should be altered. To obtain more information about the effects of the MI intervention and to verify altered muscle tone, the choice of a different measurement method (*e.g.*, Myoton)

might be beneficial. With these changes, information can be obtained on whether repeated MI training leads to increased cortical drive at rest to the muscle.

### Funding
The authors received no funding for this work.

### Competing Interests
The authors declare there are no competing interests.

### Author Contributions
- Björn Wieland conceived and designed the experiments, performed the experiments, analyzed the data, prepared figures and/or tables, authored or reviewed drafts of the article, and approved the final draft.
- Michael Behringer conceived and designed the experiments, authored or reviewed drafts of the article, and approved the final draft.
- Karen Zentgraf conceived and designed the experiments, analyzed the data, authored or reviewed drafts of the article, and approved the final draft.

### Human Ethics
The following information was supplied relating to ethical approvals ({i.e.}, approving body and any reference numbers):

The Ethics Committee of the Dept. of Psychology and Sports, Goethe University Frankfurt granted Ethical approval to carry out the study (Ethical Application Ref: 2019-63).

### Data Availability
The raw data is available in the Supplemental Files.

### Supplemental Information
Supplemental information for this article can be found online at http://dx.doi.org/10.7717/peerj.14412#supplemental-information.

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
