# Peer review of "Effects of motor imagery training on skeletal muscle contractile properties in sports science students"

_PeerJ, doi:10.7717/peerj.14412_

## Round 0.1 · original submission · Major Revisions

Thank you for submitting the manuscript to PeerJ. It has been reviewed by experts in the field and we request that you make major revisions before it is processed further.

We look forward to hearing from you soon.

Best wishes,

Badicu Georgian, Ph.D

Reviewer 1 ·

Basic reporting

First of all, I would like to thank you for the opportunity to review the research entitled "Effects of motor imagery training on skeletal muscle contractile properties". The research presents a good theoretical and methodological consistency which complies with the established research objectives and hypothesis. However, below I suggest some points for improvement, in order to improve the quality of the research:

The first one consists of structuring the abstract with the following parts: Background, Methods and Results. This way of structuring the abstract provides a better understanding of the research. It is also Peer J's policy to structure the abstract in this way.

In the introduction please add some reference to the last 5 years. Try to replace citations older than 6 years with more recent ones. In this case, if you consider it necessary, leave the older ones, but update those ideas with some new bibliographic reference less than 5 years old.

Also, add the research objective just before the study hypotheses. This objection is related to favor the reader's understanding of the research. In addition and as a recommendation, I would add a research question to complete this section.

Experimental design

The study presents a reliable and consistent experimental design, which is supported by the research objective and hypotheses. Likewise, tests validated by the scientific community and adapted to the study population were used for data collection. Another aspect that I would like to emphasize is that the research has been approved and supervised by an ethics committee.

As a suggestion to improve this section, I would add a subsection before the Study Design entitled Procedure. The purpose of this section is to tell the reader how the study participants were contacted, describing it in detail. Likewise, I would also add here the section on the ethics committee, leaving everything related to the sociodemographic characteristics of the sample, such as sex and age, in the subjects section.

Validity of the findings

The results obtained have been supported by an outstanding theoretical contextualization as well as by a reliable and ethical methodological design. I consider the data to be fully reliable and a significant advance in the field of motor imagery.

As an aspect of improvement, just after the conclusions of the study, I would add a new section entitled Limitations and Future Perspectives. In this section, the limitations of the study should be mentioned, as well as the future perspectives that the research will provide.

Additional comments

Regarding the Bibliography it is necessary to add "&" between the penultimate and last author. Also, add the entire DOI address including the following https://

·

Basic reporting

Authors reported on an interesting investigation on skeletal muscle contractile properties changing after MI practice. The manuscript is generally well-written, but there is a lack of info in introduction, rationale, objectives/hypotheses and methodology. Given the importance of the subject in the sport science field, the manuscript should give more info on the matter. Although the study results are promising and important, I have some major methodological concerns and other issues that the authors need to address before I can accept the manuscript for publication.

Specific comments

Abstract
• Authors should consider adding more info on the participants (i.e., male/female composition of the group and their status) involved in the study. Were all of them physically active people or not?
• The key words are missing.
• The title should also refer to the status of the participants (students/active).

Introduction

• The information given by the authors is not sufficient to create the background for this important matter to sport field. Can the authors maybe provide some more background on the importance of MI training? There should also be presented some info from literature or relevant studies on sport students or physically active people involved in MI training.

• The rationale for examining this problem should be mentioned more clearly in this section. Why did the authors choose to examine it? At least the reader should be given a background on how MI training is important and informed on the novelty of this study.

• Lines 138-140 are not enough to make your study of sound scientific importance. Also, the objective is not specified until line 339, in the first part of the Discussion. Please provide a subsection here with the title RESEARCH QUESTIONS/HYPOTHESES/OBJECTIVES which would be more convincing about the importance of the study.

• The last part of the Introduction is the place where some elements of the procedure are explained. The article innovation should be presented here. Describe what the research gap of the paper is and what is new. Please describe the links between the research gap and the goal of the article.

Experimental design

• The number of the participants in the study is 45, but 42 remained under evaluation. How many were female/male? Were they equally-divided in those 3 groups? Not until line 298 the reader finds out about the group composition. It is not clear how the authors formed those 3 groups. The procedure is not explained. Could this be clarified?

• It is also necessary to specify if there was a priori analysis performed to establish the sample size. This information is essential to assume the soundness of the obtained results.

Validity of the findings

• The Results section should be reorganised as to follow each hypothesis/research question or objective. Authors need to write key findings focusing on each one of these after being stated.

• The Discussion section - first paragraph should also refer to what is new regarding the findings of the study.

• It would be better to have seen more use of terms like 'originality' and 'significance'. Identify what is new in this study that may benefit readers or how it may advance existing knowledge or create new knowledge on this subject. There should be a clear conclusion on why the research findings are significant for sport field.

• It seems that the English is clear, but research articles usually do not use the word "we/our" and regularly use passive verbs.

• Research limitations and existing problems are not distinctly presented.

• A list of abbreviation would be helpful.

Reviewer 3 ·

Basic reporting

The English language should be improved in some points.

The introduction of literatures is well written, but it needs to be slightly shorten, e.g. paragraph 2-3 (the research done in MI on corticospinal level which fully not investigated much in the present study)

Thanks for your shared raw data. Table 1 may be considerate to remove since it's already explained in Fig 3-5.

Experimental design

The research question is well addressed and fitted to the scope of the J that there was no study on muscle contractility in long term MI intervention.

Methodology is also well defined but it needs to be reorganize to better follow and understanding. Please separate between study design and measurement procedures. The order of variable testing, the tested side of biceps brachii, how to allocate participants' group, number of groups, the number of training session should be better to state in study design part. The measurements of MVC, imagery ability, and TMG should be define as measurement subsection.

Please clarify in details of number of testing for MVC and TMG each conditions.

Please also clarify whether all measurements were made by the same investigators and the room temperature was well controlled for all testing sessions.

Validity of the findings

Conclusions are linked to the original research question, but slightly needs to be improved. The summation of MVC result should also be defined here. Line 441-448 should be remove from conclusion, hence the recent finding and purpose of this study should not be replicated here.

Additional comments

Thank you for your work and interesting finding. The following recommendations are suggested to improve this MS.

Abstract; It's not necessary to report F test here, only p value and partial eta squared enough.

Introduction;
Line 43 should be (EMG) (Stinear.....)
Line 46 please remove (for a review, see)
Line 119 please remove - BMC, Ljubljana, Slovenia

Method;
Line 161 please clarify why the participants were asked to maintain their regular activities, which would not affect the result or interpretation of the present study when they were trained?
Line 227-228 Please remove More details for each...

Statistical; Was there any data distribution test before using ANOVA?

Results;
Please remove all "see" before Fig.
It would be better if number of males in Line 298-299
Line 322 should be no time effect,
Line 323 should be group effect, or followed by F test in parenthesis, please also check the similar issue on Line 328-332

Discussion;
This part is well stated and having a supportive scientific evidences and possible explanations for all study variables.
The end of the first Paragraph regarding MVC, it would be better to compare the relative strength or raw MVC of baseline in the present to other study to clarify their fitness muscular state.
Line 391-392 Please revise the sentence
Line 394 please change "symbol of change" into the text and also clarify the two r reported in each valve what variable associations, this also in Line 397-399
Line 431-438 or last paragraph should be whether concerned as a limitation part of this study?

---

## Round 0.2 · accepted · Accept

Thank you for addressing the minor revisions. The manuscript can now be accepted for publication.

Reviewer 1 ·

Basic reporting

Having reviewed the new version of the article, I believe that it meets the criteria and quality of this journal.

Experimental design

The experimental design is very well justified and all the established steps have been followed.

Validity of the findings

The results are very interesting and clearly explained. They bring a great scientific novelty.

·

Basic reporting

Thank you for providing this comprehensive work.
The authors have presented an improved version of the manuscript.

The introduction provides a proper background of the topic. The sections and the title have been improved. Relevant results are well-organized to follow the hypothesis.
The quality of the images is good enough.
It seems that the English is technically correct.

Experimental design

The experimental design meets the scope of the journal, and it is relevant to the community.
Methods are described detailed enough.

Validity of the findings

The results and the conclusions are quite interesting and well-discussed. All data are provided.

The authors have adequately addressed all my comments. I have no further suggestions.

Reviewer 3 ·

Basic reporting

The suggestions have been well addressed throughout the MS. Thank you very much for your revise.

Experimental design

no comment

Validity of the findings

no comment

Additional comments

no comment